# Modeling hereditary diffuse leukoencephalopathy with axonal spheroids using microglia-sufficient brain organoids

**Wei Jie Wong[1†], Yi Wen Zhu[1†], Hai Ting Wang[1], Jia Wen Qian[1], Ziyi Li[1], Song Li[1], Zhao Yuan Liu[1], Wei Guo[1], Shuang Yan Zhang[1], Bing Su[1], Fang Ping He[2], Kang Wang[2]\*, Florent Ginhoux[1,3,4,5,6]\***

[1]Shanghai Institute of Immunology, Department of Immunology and Microbiology, Shanghai Jiao Tong University School of Medicine, Shanghai, China; [2]Department of Neurology, First Affiliated Hospital, School of Medicine, Zhejiang University, Hangzhou, China; [3]Singapore Immunology Network, Agency for Science, Technology and Research, Singapore, Singapore; [4]Department of Microbiology and Immunology, Yong Loo Lin School of Medicine, National University of Singapore, Singapore, Singapore; [5]INSERM U1015, Gustave Roussy Cancer Campus, Villejuif, France; [6]Translational Immunology Institute, SingHealth Duke-NUS Academic Medical Centre, Singapore, Singapore

**\*For correspondence:**
fcwangk1@zju.edu.cn (KW);
florent_ginhoux@immunol.a-star.edu.sg (FG)

†These authors contributed equally to this work

## eLife assessment

This study presents **valuable** findings on the mechanisms underlying a rare brain disease using an organoid system. In this revised version, there are remaining reviewers' comments that are not yet addressed and as such, while the data presented are **solid**, the evidence supporting some of the claims is deemed incomplete. The work will be of interest to neuroscientists and clinicians aiming to understand and combat similar neurodegenerative disorders.

**Abstract** Hereditary diffuse leukoencephalopathy with axonal spheroids (HDLS) is a rare, fatal, adult-onset neurodegenerative disease that is most often caused by mutations affecting the colony stimulating factor-1 receptor (CSF-1R). To understand how CSF-1R-mutation affects human microglia – the specialized brain-resident macrophages of the central nervous system – and the downstream consequences for neuronal cells, we used a macrophage and forebrain organoid co-culture system based on induced pluripotent stem cells generated from two patients with HDLS, with *CSF-1R* gene-corrected isogenic organoids as controls. Macrophages derived from iPSC (iMacs) of patients exhibited a metabolic shift toward the glycolytic pathway and reduced CSF-1 sensitivity, which was associated with higher levels of IL-1β production and an activated inflammatory phenotype. Bulk RNA sequencing revealed that iMacs adopt a reactive state that leads to impaired regulation of neuronal cell populations in organoid cultures, thereby identifying microglial dysregulation and specifically IL-1β production as key contributors to the degenerative neuro-environment in HDLS.

## Introduction

Hereditary diffused leukoencephalopathy with axonal spheroids (HDLS) – also known as adult-onset leukodystrophy with axonal spheroids and pigmented glia (ALSP), is a rare, inherited, autosomal dominant, neurodegenerative disease that is characterized by patchy axonal swellings (spheroids) and demyelination that result in the alteration of the white matter of the brain (*Marotti et al., 2004*). Although detailed epidemiological data are lacking, in 2021, it was estimated that approximately a quarter of a million people globally are affected by the condition (*Papapetropoulos et al., 2021*); yet our understanding of the pathogenesis of the disease is limited, and there is no cure or even effective treatment.

For more than a decade, we have known that HDLS is caused by mutations affecting the colony-stimulating factor-1 receptor (CSF-1R) (*Rademakers et al., 2011*) which most often occur in the tyrosine kinase domain (TKD) (*Zhuang et al., 2020*), and lead to reduced auto-phosphorylation in response to CSF-1, which results in impaired downstream signaling (*Zhuang et al., 2020*; *Pridans et al., 2013*). These mutations most profoundly affect microglia (*Biundo et al., 2021*), the specialized brain-resident macrophages that play critical roles in brain development and homeostasis (*Tay et al., 2017*; *Hammond et al., 2018*) as well as in a wide range of neuroinflammatory and neurodegenerative diseases (*Bachiller et al., 2018*). Patients with HDLS have few, small, microglia that are distributed abnormally within the neocortex and that express low levels of the homeostatic marker P2RY12 and high levels of the inflammatory marker CD68, compared to healthy controls (*Kempthorne et al., 2020*; *Konno et al., 2018*; *Tada et al., 2016*).

Studies aiming to understand the pathophysiology of HDLS have used various approaches. Rodent models have been generated, but do not faithfully recapitulate the disease phenotype in humans: the earliest haplo-insufficient $Csf-1r^{+/-}$ mice model exhibited microgliosis within brain regions (*Chitu et al., 2015*), but this was not recapitulated in a later Csf-1r knockout rat model (*Patkar et al., 2021*) while a subsequent orthologous mouse model has phenocopied the reduced microglia density seen in patients with HDLS but does not exhibit brain pathology (*Stables et al., 2022*). A zebrafish model of HDLS similarly saw reduced microglia during brain development (*Berdowski et al., 2022*), but does not offer the opportunity to study pathology associated with adult-onset neurodegenerative diseases (*Wang et al., 2021*). To overcome these species-specific limitations, in this study we generated human induced pluripotent stem cell (iPSC)-derived macrophages from patients with HDLS and used them to create microglia-sufficient autologous brain organoids (*Park et al., 2023*) in which we characterized the effects of naturally occurring *CSF-1R* mutation on microglial function. By comparing iPSC-derived macrophages to genetically corrected macrophages from the same patients, we uncovered evidence of significant transcriptional and metabolic reprogramming that was associated with profound changes to neuronal regulation by these cells, indicative of the likely mechanistic links between microglial dysfunction and HDLS.

## Results

### Generation and characterization of iPSC lines from patients with HDLS

We first isolated dermal fibroblasts from skin biopsies taken from two patients with HDLS and confirmed mutations in the TDK region of the *CSF-1R* gene (HD1, HD2; clinical characteristics shown in *Supplementary file 1, table S1*) then cultured them for approximately 4 weeks before reprogramming into induced pluripotent stem cells (iPSCs) (*Figure 1A*). Four to five weeks post-transfection, we observed iPSC-like colonies emerging (*Figure 1—figure supplement 1B*), and 7–10 days later, we selected those clones that were proliferating and displayed typical iPSC morphology (*Figure 1C*). We confirmed successful reprogramming by visualization of pluripotency markers including SOX2 and OCT4 (nuclear markers), and TRA-1-60 and SSEA-4 (intracellular marker), by immunofluorescence (*Figure 1D*). We also confirmed that these clones had normal karyotype (*Figure 1E*) and termed them Mut HD1 and Mut HD2. One clone per cell line was used for further analysis. We then generated isogenic from Mut HD1 and Mut HD2 iPSC cell lines which we termed IsoHD1 and IsoHD2, respectively, as controls (see Methods). Clones that were karyotypically normal were used for downstream experiments alongside their mutant counterparts (*Figure 1B*), with no off-target events identified (*Figure 1—figure supplement 1C*). Collectively, HDLS iPSC and their corresponding isogenic iPSC controls were successfully generated from two patients with HDLS (HD1 and HD2).

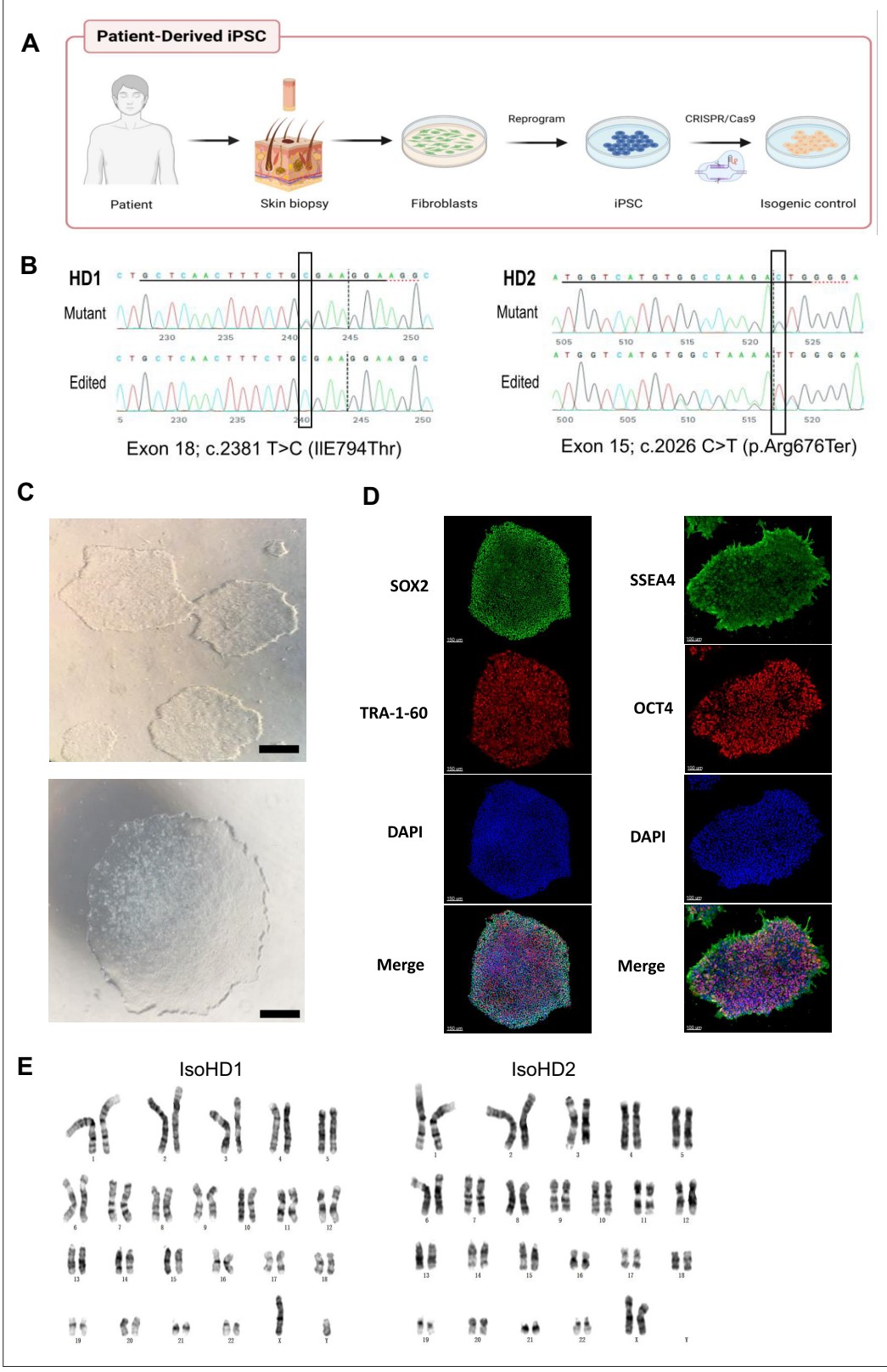

**Figure 1.** Generation and characterization of iPSC lines from patients with HDLS. (**A**) Schematic representation of the generation of patient-derived induced pluripotent stem cells (iPSC) from dermal fibroblasts, and the subsequent generation of isogenic controls via gene editing with CRISPR/Cas9. (**B**) Sequencing of the genomic CSF-1R locus in donor-derived iPSC displaying mutant allele or genetically edited allele in isogenic control cell

*Figure 1 continued on next page*

*Figure 1 continued*

lines. (**C**) Phase-contrast microscopy of mature reprogrammed iPSC colonies displaying typical hESC morphology. Images were acquired with a standard microscope (Nikon) with a 10× objective. Scale bar represents 250 μM. (**D**) Immunofluorescence imaging of Mut HD1 iPSC colonies for the pluripotency markers SOX2, OCT4, TRA-1-60, and SSEA-4. Scale bar represents 150 and 100 μM for left and right panels, respectively. (**E**) Cytogenetic analysis of IsoHD1 (left) and IsoHD2 (right) iPSC clone showing a normal karyotype.

The online version of this article includes the following figure supplement(s) for figure 1:

**Figure supplement 1.** Donor-derived human dermal fibroblast cells and induced pluripotent stem cells (iPSC) and off-target events (OTE) analysis.

## Generation and characterization of patient iPSC-derived macrophages

We next sought to differentiate primitive microglia-like macrophages from patient-derived iPSC cell lines (iMacs) using our serum-free protocol that faithfully recapitulates the generation of in vivo microglial progenitor cells (*Takata et al., 2017*). Briefly, iPSCs were initially inducted toward the meso-dermal lineage before being differentiated into hemangioblast-like cells with the potential to differ-entiate into both endothelial and hematopoietic progenitors – the latter emerging as free-floating cells as early as day 6 after seeding – which we collected and then terminally differentiated into iMacs by the addition of CSF-1 (*Figure 2A*). Although HDLS mutation downregulates the sensitivity of CSF-1R to its ligand (*Pridans et al., 2013*), we were able to successfully generate iMacs using this protocol: the yield of iMacs from iPSC varied between cultures (*Figure 2B*), but flow cytometry confirmed their consistent macrophage-like phenotype, with abundant expression of CD45 and CD14 (*Figure 2C*), and Giemsa staining showed that iMacs exhibited comparable morphology between donors (*Figure 2D*). Functionally, all iMac cultures were equally able to take up PE-labeled beads that were detectable within their cell bodies (*Figure 2D*); flow cytometry confirmed that the level of actin-dependent phagocytosis of pHrodo beads was comparable among CD45-expressing cells from all cultures (*Figure 2E*). Collectively, these data show that iMacs generated from HDLS donor-derived iPSC adopted the typical morphology, phenotype and functions associated with macrophages.

The mutations associated with HDLS downregulate the sensitivity of CSF-1R to its ligand CSF-1, but to different extents depending on the mutation site (*Pridans et al., 2013*). We therefore compared the survival of our mutant and isogenic cells lines from both patients during culture with CSF-1. As expected, even with high concentrations of CSF-1, relatively few mutant cells survived across 7 days of culture; by contrast, even low CSF-1 levels allowed significantly more isogenic cells to survive (*Figure 2F*).

In summary, we generated iMacs from iPSC of patients with HDLS and their respective isogenic controls that exhibit typical macrophage morphology, phenotype, and phagocytic capabilities.

## Mutant and isogenic control iMacs have distinct gene expression profiles

Although iMacs derived from patients with HDLS exhibited core macrophage features, we wanted to more fully understand the impact of *CSF-1R* mutation on these cells. We first characterized their gene expression profile using bulk RNA-seq and compared it to that of their respective *CSF-1R*-restored isogenic iMac cultures.

Principal component analysis (PCA) showed that patterns of gene expression differed markedly between mutant and isogenic iMacs. Among cell cultures from HD1, mutant iMacs exhibited patterns of gene expression consistent with a more reactive state compared to isogenic *CSF-1R*-restored HD1: HD1 Mut had higher expression of genes involved in pathways related to proliferation (IGF2, MYBL2, STAT1, INCEP, TGFA) and the defense response to virus (IFI27, IFI44L, STAT1, MX1, IFIT3); while Iso HD1 iMacs expressed high levels of genes involved in chemokine-related responses and signaling pathways (CXCL2, ADAM8, CCL31L, CCL5), as well as leukocyte migration and chemotaxis (CXCL3, CCL13, CCL2, E5AR1, C3AR1) (*Figure 3A–C*). We observed a similar transcriptomic profile in HD2-derived iMacs: HD2 Mut showed upregulated gene expression for the defensive response to virus, while isogenic HD2 iMacs exhibited high expression levels of genes involved in leukocyte migration and chemotaxis (*Figure 3D–F*). These data are in line with previous reports suggesting that macro-phages from patients with HDLS adopt an inflammatory phenotype with increased expression of the

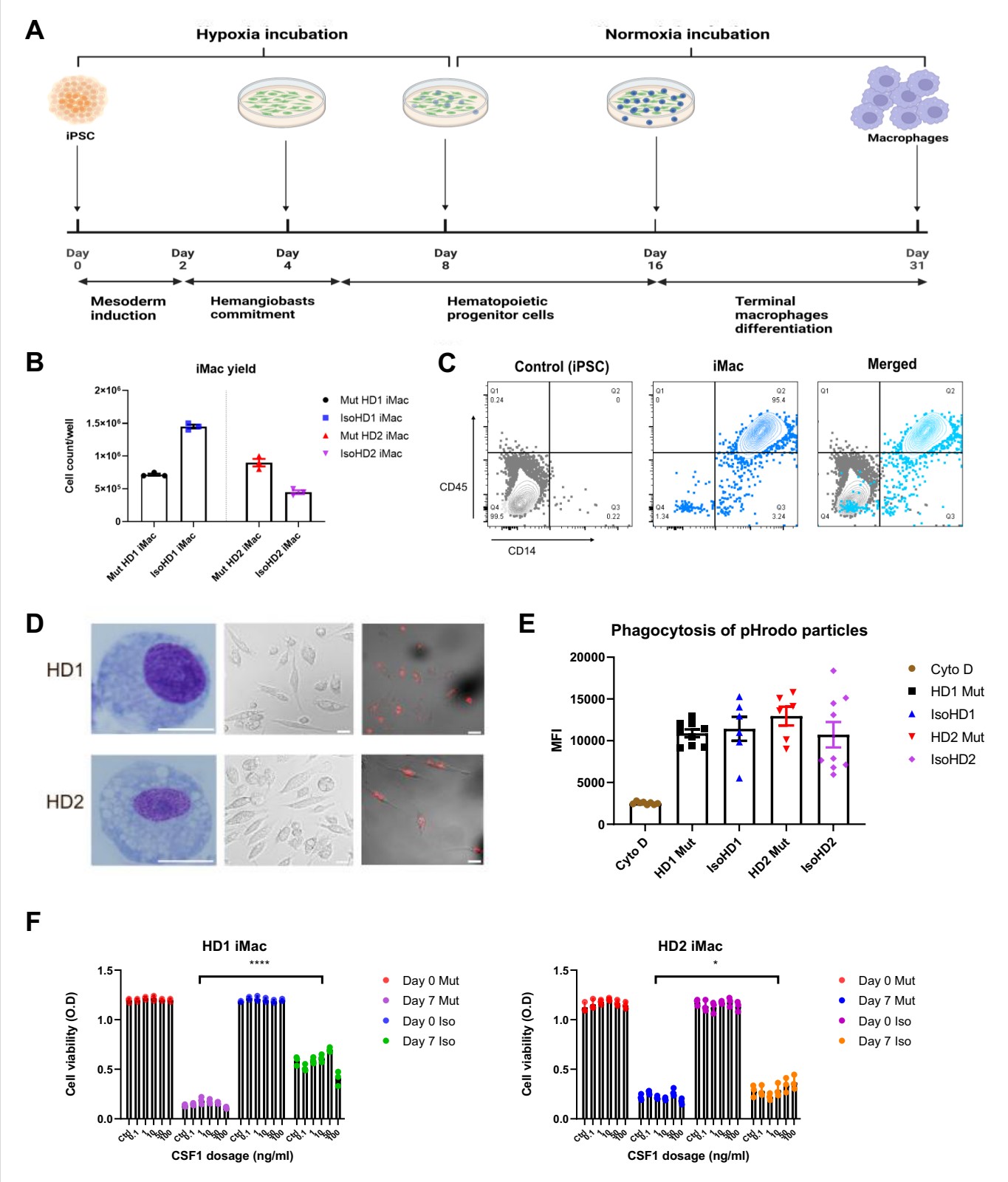

**Figure 2.** Generation and characterization of patient iPSC-derived macrophages. (**A**) Schematic representation of the derivation of primitive macrophages from donor-derived iPSC. (**B**) iMacrophage (iMac) yields across all cell line variants. (**C**) Flow cytometry analysis of iPSC and iMac expression of macrophage surface markers CD45 and CD14. (**D**) Morphology of iMacs visualized by Giemsa staining (left panel), and confocal imaging (middle panel); and internalization of PE-conjugated beads visualized by confocal imaging. Scale bar represents 30 µM. (**E**) Uptake of pHrodo-labeled

*Figure 2 continued on next page*

*Figure 2 continued*

beads by iMacs in the presence and absence of cytochalasin D (Cyto D). (**F**) Cell viability of iMacs used to determine CSF-1 sensitivity via an MTT assay, after incubating with varying concentrations of CSF-1 for 7 days. Data are presented as mean ± SEM. Statistical significance was assigned as: p < 0.05 was considered significant: *p < 0.05; **p < 0.01; ***p < 0.001; ****p < 0.0001.

immunoreactive markers CD68 and CD163 (*Kempthorne et al., 2020*; *Tada et al., 2016*). Additionally, an upregulation of the immature myeloid cell marker MPO (*Pinkus and Pinkus, 1991*), and a reduced expression of mature macrophage marker MARCO (*Elchaninov et al., 2021*) was observed within mutant iMac of both cell lines. Taken together, iMac cultures derived from patients with HDLS exhibited transcriptomic profiles consistent with upregulated inflammatory gene expression as a result of impeded CSF-1R function.

## Mutant and isogenic control iMacs have distinct metabolic profiles

Studies have shown that activated macrophages often undergo metabolic reprogramming to fulfil the energetic needs of demanding biochemical processes (*Kolliniati et al., 2022*; *Jha et al., 2015*); alongside, we know that energy metabolism, as well as metabolic cell features, plays an important role in mediating macrophage function and plasticity (*Xue et al., 2014*). Therefore, we next compared the cellular metabolic profiles of patient-derived *CSF-1R* mutant and isogenic *CSF-1R*-restored iMacs.

We first assessed glycolysis by measuring the extracellular acidification rate (ECAR) in the presence of specific modulators (glucose, oligomycin, 2-deoxy-glucose) that were added sequentially to the cell cultures to reveal various aspects of the metabolic pathway (see Methods): this showed that *CSF-1R*-mutant iMacs from both HD1 and HD2 had a significant overall increase in glycolytic function (*Figure 3G, O*). Although the magnitude of the difference between iMac HD1 and HD2 and their respective *CSF-1R*-restored counterparts varied, in both cases, *CSF-1R* mutation was associated with significantly increased glycolytic capacity and glycolytic reserve.

Next, we determined the functional metabolic profile of mitochondrial respiration by measuring real-time changes in the extracellular oxygen consumption rate (OCR) during sequential treatment of cells with oligomycin, carbonyl cynide *p*-trifluoromethoxyphenylhydrazone (FCCP) and a combination of rotenone and antimycin A. As before, although the magnitude of effect was less for cells from HD1, both sets of iMacs showed a similar trend toward higher OCR values, and greater capacity for basal respiration, ATP production, and maximal respiratory capacity in the presence of mutated *CSF-1R* in the case of HD2 Mut cells, these differences achieved statistical significance across all three bioenergetic parameters.

Taken together, these results indicate that the absence of fully functional CSF-1R in cells from HD1 and HD2 is associated with an upregulated glycolytic metabolic profile, which is cells from HD2 is clearly accompanied by a higher mitochondrial oxidative phosphorylation rate. This suggests that macrophages from patients with HDLS have an altered metabolic profile that correlates with their higher activation status and could contribute to disease pathology.

## HDLS iMacs upregulate IL-1β when exposed to apoptotic neuronal cells

Our data suggest that mutant iMacs have dysregulated transcriptomic and metabolic profiles, suggestive of an active inflammatory state. To better understand how this would be expressed in the context of their functions in the brain, we mimicked a need for apoptotic cell clearance by incubating iMacs from patients with HDLS with UV-irradiated neuronal SH-SY5Y cells (*Figure 4—figure supplement 1A*). Both iMac lines (*CSF-1R* mutant and isogenic) from both patients effectively phagocytosed apoptotic but not control, non-UV-irradiated, SH-SY5Y cells, in an actin-dependent manner (*Figure 4—figure supplement 1B*).

We then measured the expression of genes encoding pro- (IFN-g, TNF-α, IL-1β, IL-6, IL-18) and anti- (IL-10, IL-12, TGF-β) inflammatory cytokines following incubation of iMacs with UV-irradiated or non-irradiated SH-SY5Y cells, or without any SH-SY5Y cells, for 8 hr. We observed a significant and specific increase in IL-1β transcription in *CSF-1R* mutated iMacs from both HD1 and HD2 when exposed to apoptotic cells, compared with their isogenic counterparts. Among the anti-inflammatory cytokines, Mut HD2 cells exhibited significantly increased transcription of the gene encoding TGF-β when co-cultured with UV-treated cells, alongside high baseline levels of IL-12 and TGF-β in control cultures compared to isogenic HD2 iMacs. Interestingly, all *CSF-1R* mutant iMac groups had higher

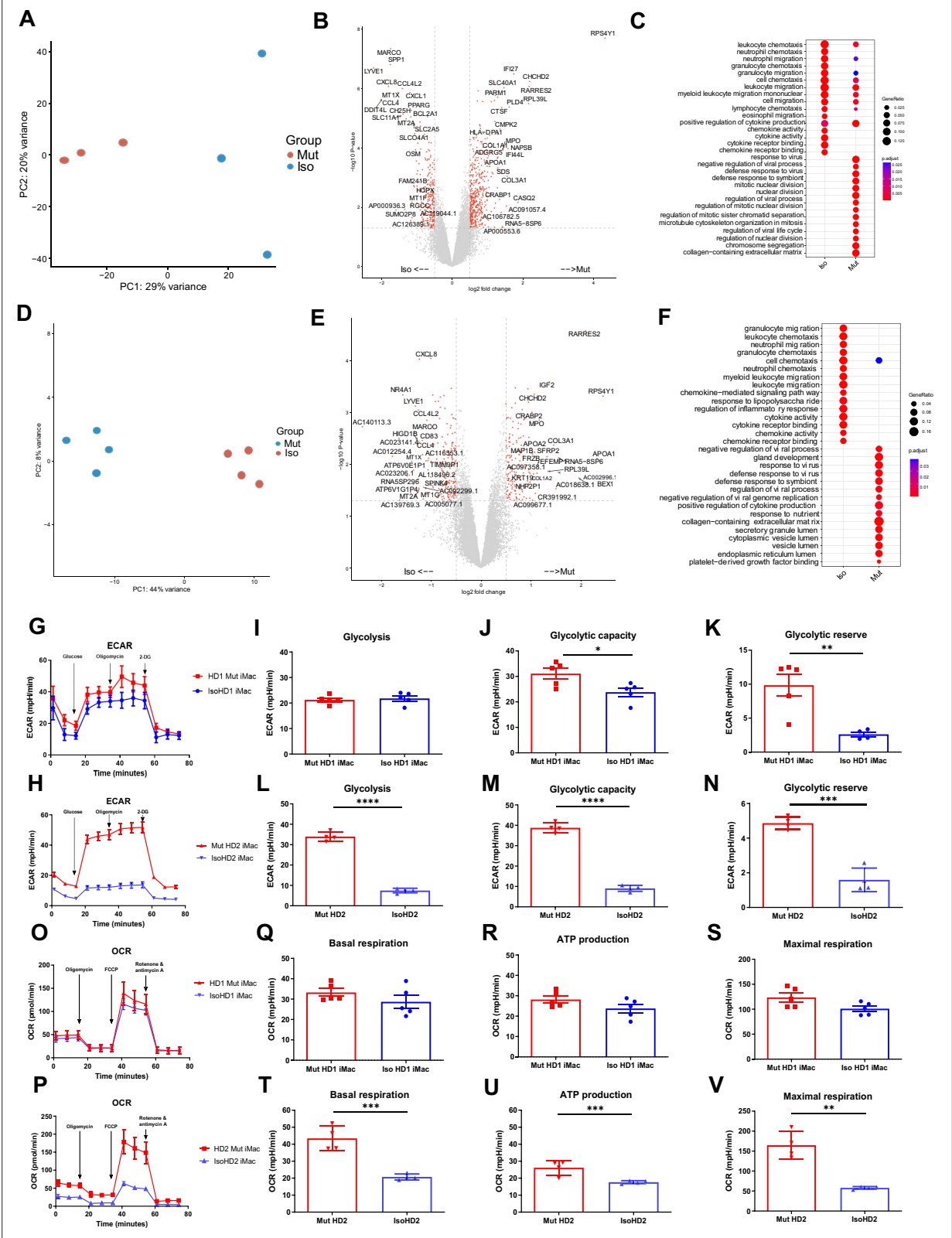

**Figure 3.** Mutant and isogenic control iMacs have distinct gene expression profiles. (**A, D**) Principal component analysis (PCA), (**B, E**) volcano plots, and (**C, F**) GO analysis, of mutant and isogenic iMacs derived from HD1 and HD2 differentially expressed genes (DEGs), respectively. Representative curve of extracellular acidification rate (ECAR) (**G, O**) and their respective quantified data (**H–J, P–R**) are shown. Representative curve of oxygen consumption rate (OCR) (**K, S**) and their respective quantified data (**L–N, T–V**) are shown. Real-time assessment of bioenergetic profile between mutant and isogenic

*Figure 3 continued*

HD1 iMacs, measured using extracellular flux assay and a mitochondrial stress test. The following glycolytic parameters were calculated based on ECAR: (**B**) glycolysis, (**C**) glycolytic capacity, and (**D**) glycolytic reserve. Data shown as means + SEM, *n* = 9. The following parameters were calculated based on OCR: (**F**) basal, (**G**) ATP production, and (**H**) maximal respiration. 2-DG = 2-deoxy-glucose, FCCP = carbonyl cyanide 4-trifluoromethoxy-phenylhydrazone (*n* = 3). Data are presented as mean ± SEM. Statistical significance was assigned as: p < 0.05 was considered significant: *p < 0.05; **p < 0.01; ***p < 0.001; ****p < 0.0001.

levels of transcription of most pro-inflammatory cytokines at baseline, consistent with their activated/reactive transcriptional profile, than did their *CSF-1R*-restored counterparts (*Figure 4A, B*).

Collectively, all iMacs exhibited the ability to specifically phagocytose apoptotic cells, but in the absence of a fully functional CSF-1R, this led to high levels of IL-1β transcription, which reflected a general propensity toward the transcription of pro-inflammatory cytokine genes, even in the absence of overt stimulation. Importantly, restoration of *CSF-1R* via gene editing was sufficient to reverse this inflammatory phenotype.

## iMacs differentiate into microglia-like cells (iMicro) in a co-culture with forebrain organoid

During HDLS, patients typically exhibit degeneration of the frontal lobes of the brain, which contributes to the deficit and decline in cognitive ability, changes to personality and behavior, and dysfunctional

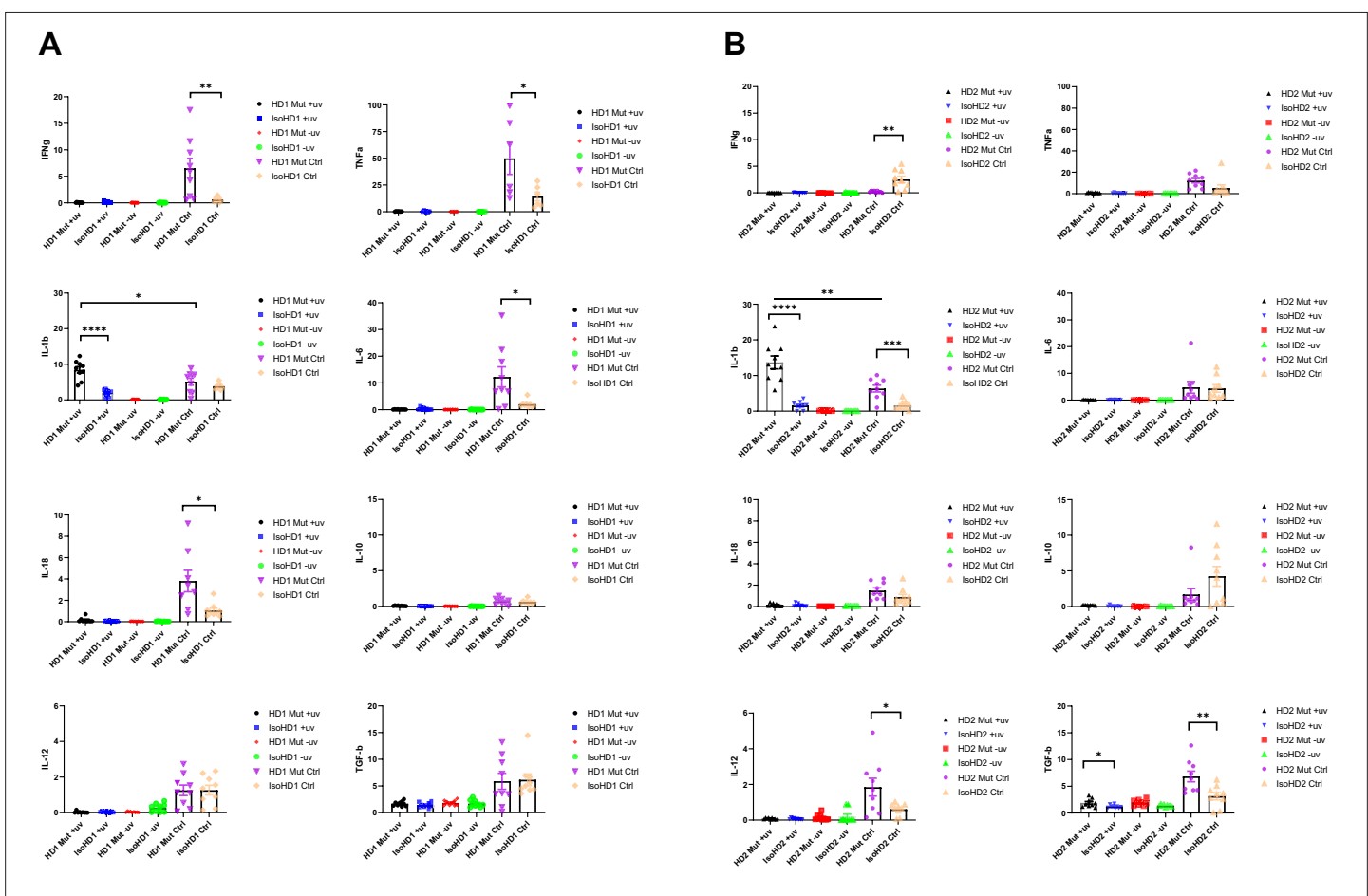

**Figure 4.** Cytokine profiling of iMacs saw upregulated levels of IL-1β when probed with apoptotic neuronal cells. (**A, B**) Levels of transcription of pro- and anti-inflammatory cytokine genes (IFNg, TNFa, IL-1b, IL-6, IL-18, IL-10, IL-12, and TGFb) quantified by qPCR (+uv depicts UV-treated, −uv depicts non-UV treated). *n* = 3 per cell line. Data are presented as mean ± SEM.

The online version of this article includes the following figure supplement(s) for figure 4:

**Figure supplement 1.** Characterization of apoptotic SH-SY5Y neuronal cells.

motor, social, and language skills (*Freeman et al., 2009*; *Sundal et al., 2012*). Therefore, we next generated iPSC-derived forebrain organoids including autologous iMacs from patients with HDLS to better understand their potential role in the brain.

To better understand how diseased or non-diseased macrophages affect the developing brain, we first generated forebrain organoids from the isogenic cell line variant of the respective donor's iPSC (see Methods). After 31 days, organoids were harvested (*Figure 5—figure supplement 1A*) and co-cultured for 7 days with isogenic or mutant iMacs that had been generated in parallel (*Figure 5A*). At this point, each organoid was removed from its co-culture and cultured individually for 16 more days (*Figure 5B*).

We then applied 3D imaging of forebrain organoids to assess their morphology and the distribution/differentiation of the iMacs. This showed that iMacs expressing the microglial marker IBA-1, were mostly located on the surface of the organoids (*Figure 5C*). We also labeled organoids for NESTIN (a neuronal cell marker) and SOX2 expressed by neuronal progenitor cells (NPCs). In all of the organoids, neural epithelial rosettes were present, consisting mostly of SOX2$^+$ NPCs (*Figure 5—figure supplement 1B*); these structures indicate favorable organoid growth as they demonstrate proper lineage progression leading to forebrain region identity (*Lancaster et al., 2013*) Importantly, organoids to which we did not add iMacs did not contain any IBA-1$^+$ cells (*Figure 5—figure supplement 1C*); this was as expected from our previous work (*Park et al., 2023*), but was in contrast to previous studies that reported innately derived microglia in mesodermal-lineage-containing cerebral organoids (*Lancaster et al., 2013*; *Ormel et al., 2018*).

Next, we characterized the different cell populations within organoids after 23 days of co-culture and compared iMacs before and after co-culture. After organoid dissociation, cells were labeled with antibodies to differentiate neurons (CD45$^-$/CD184$^-$/CD44$^-$/CD15$^{lo}$/CD24$^+$), NPCs (CD45$^-$/CD184$^+$/CD271$^-$/CD44$^-$/CD24$^+$), and iMacs (CD45$^+$) before undergoing sorting (*Figure 5D*) followed by bulk RNA sequencing. During the course of the work, sufficient iMicro numbers for downstream analysis were only generated from co-culture with mutant iMacs for HD1 and isogenic iMacs for HD2, respectively. Compared to day 31 non-co-cultured iMacs, HD1 iMacs after co-culture had profoundly altered their gene expression (*Figure 5E*), with the upregulated DEGs showing an increase in various pathways that are characteristic of microglial functions in the brain such as central nervous system and brain development (CNTN1, ZIC2, EFNA2, GLI3), synapse organization, synapse and neuron organization and development (SLC1A1, DGKB, TRIM67), as well as neurogenesis (CHD11, ADCYAP1) (*Figure 5F, G*).

Likewise, for HD2 iMacs, PCA also suggested that their patterns of gene expression were markedly different between co-cultured and non-co-cultured populations (*Figure 5H*), while GO analysis of the upregulated DEGs revealed an increase in various pathways linked with typical microglial functions in the brain such as glial cell differentiation and gliogenesis (CLU, MDK, CXCR4, RELN, TMEM98, TUBA1A), axonal development (UCHL1, MAP1B, CRABP2, TUBB2B, LGI1) as well as forebrain development (TUBB2B, MDK, CXCR4, RELN, LHX2) (*Figure 5I, J*). This shows that iMacs undergoing co-culture with forebrain organoids have differentiated to a certain extent, into microglia-like cells.

## Co-culture results in impaired regulation of neurons within organoids

Microglia not only act as sentinel cells, but are also important contributors to neuroplasticity and neurogenesis in the developing brain that ultimately help shape the brain circuitry (*Wake et al., 2011*). Part of this role requires microglia to regulate the population of NPCs and neurons during the unrestrained, highly proliferative phase of early brain development (*Cunningham et al., 2013*; *Sierra et al., 2010*); therefore, we next asked whether the microglia-like cells from our iMac-forebrain organoids were also capable of regulating neurogenesis in vitro.

As such, we sought to compare the size of the organoids, as well as the individual major cell population in each organoid, for both variants between isogenic organoids that did not undergo co-culture, and those that underwent 23 days of co-culture by comparing their area before and after co-culture. Co-culture of HD1 saw a slight reduction in the organoid size among both co-culture parameters before and after co-culture. However, comparison of the organoid size on co-culture day 23 only, saw the isogenic control parameter as the only group that had a reduction in organoid size when compared to the organoid-only control (*Figure 6A*). On the contrary, HD2 displayed no significance in organoid size reduction even after co-culture (*Figure 6B*).

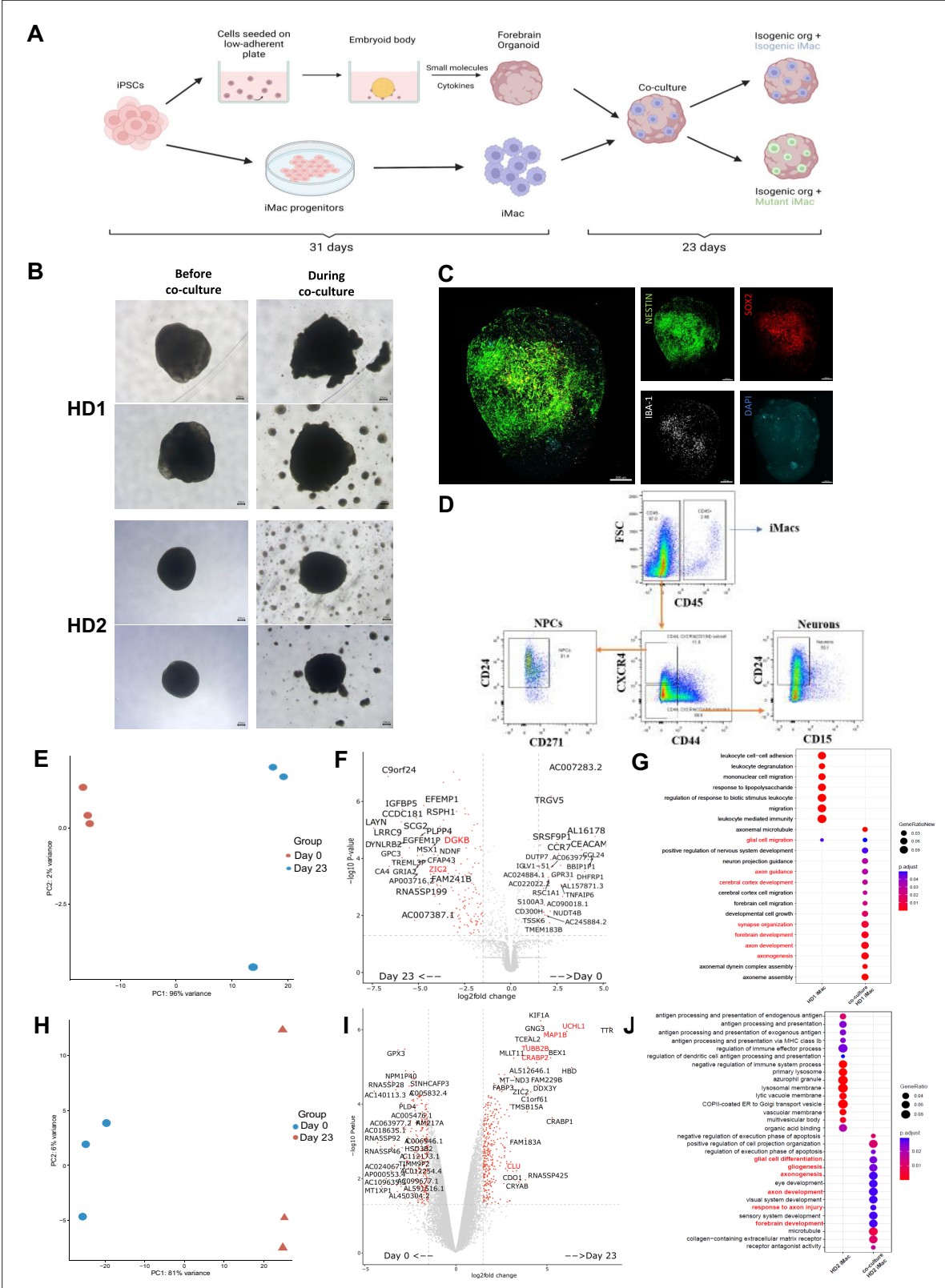

**Figure 5.** iMacs differentiate into microglia-like cells in co-culture with forebrain organoids. (**A**) Schematic diagram of co-culture between day 31 iMacs and forebrain organoid. (**B**) Forebrain organoids depicted before, during, and after co-culture with iMacs. Day 31 isogenic forebrain organoids derived from the respective donor iPSC cell line were co-cultured with the same isogenic- and mutant-cell-line-derived day 31 iMacs. 30,000 iMacs were co-cultured with each respective organoid derived from HD1 and HD2. Figure shows matured organoids before and after the addition of iMacs during

*Figure 5 continued on next page*

*Figure 5 continued*

co-culture. (**C**) Three-dimensional visualization of co-cultured brain organoids after 23 days. Neurons were labeled for NESTIN and NPCs for SOX2. Microglia-like cells (IBA-1+) were visualized on the surface of organoids. (**D**) Flow cytometry analysis of neurons, NPCs, and macrophage populations within forebrain organoids after 23 days of co-culture. (**E, F**) Donor-derived iMacs in forebrain co-cultures undergo distinct changes in gene expression as observed in principal component analysis (PCA) (**E, H**), volcano plots (**F, I**), and GO analysis (**G, J**) in both HD1 and HD2 variants. Scale bar represents 200 μM.

The online version of this article includes the following figure supplement(s) for figure 5:

**Figure supplement 1.** Phenotypic characterization of organoids and iMicro.

Subsequently, these co-cultured organoids were dissociated, stained, sorted, and quantified for the respective cell populations. Co-culture of IsoHD1 organoid with IsoHD1 iMac saw a significant reduction in the neuronal population when compared to control organoids that did not underwent co-culture. Conversely, co-culture with mutant HD1 iMac saw a smaller degree of neurons reduction but was not significant (*Figure 6C*). Next, co-culture of IsoHD2 organoid with IsoHD2 iMac saw a slight reduction in neuronal population that was not statistically significant, while the co-culture with mutant HD2 iMac saw no difference when compared with control organoids (*Figure 6D*). Interestingly, we did not detect any change in the NPC population for either HD1 or HD2 after co-culture, likely because their numbers were relatively low to begin with due to the cellular composition of the forebrain organoid. Similarly, the iMac population was also small in all co-cultures. Transcriptomic analysis of the neurons of HD2 co-culture saw the mutant iMac co-culture having upregulated expression of genes associated with mitochondrial respiration (BCS1L, MRPL18, TSFM, MRPL11, MRPS30) but this phenomenon was not present in the isogenic iMac co-culture (*Figure 6E*). Conversely, transcriptomic analysis of the NPCs of HD2 co-culture saw the isogenic iMac co-culture having upregulated expression of genes associated with mitochondrial respiration (*Figure 6F*). Taken together, our model suggests that HDLS mutation affects neuronal population in a microglia-dependent manner and that this phenomenon is variable and mutation dependent.

## Discussion

HDLS is an incurable neurodegenerative disease that begins in adulthood, progresses rapidly and has a grim prognosis. Efforts to elucidate the functional or dysfunctional aspects of the causative mutations in the *CSF-1R* gene have been hampered by the lack of an accurate disease model that recapitulates the human pathology: although rodent and zebrafish models have been generated and imparted valuable insights toward HDLS (*Chitu et al., 2015*; *Patkar et al., 2021*; *Stables et al., 2022*; *Berdowski et al., 2022*), discrepancies and limitations in each case have led to incomplete recapitulation of the disease's pathology, specifically in microglia numbers within brain regions and the effects on respective cognitive and sensorimotor tests. Additionally, a model looking into immune-neuronal crosstalk to better understand a neurodegenerative disorder that is primarily associated with microgliopathy, is important to advance the field. Here, we first generated iPSC lines from two patients with HDLS and then used CRISPR/Cas9 technology to reinstate a fully functional *CSF-1R* gene into some of the lines. While the generation of an HDLS donor-derived iPSC cell line has been reported (*Wu et al., 2021*), no other in vitro studies have yet reverted the causative mutation to wild type; a necessary step to understanding the direct impacts of diminished/absent CSF-1R signaling.

We then generated and characterized iMacs – precursors of microglia – from these lines, which exhibited typical morphology, phenotype, and phagocytic capacity that were not obviously affected by *CSF-1R* mutation. Although we were able to generate phenotypically and functionally similar iMac across these lines for example for phagocytic capability, iMac yield varied greatly between the cell lines with a 50% lower yield for Mut HD1 compared to IsoHD1 and, conversely, a higher yield for Mut HD2 200% compared to IsoHD2. This fundamental difference could be attributed to their respective mutations: the nonsense mutation identified in HD2 could not only more profoundly affect receptor signaling, but might also be a major factor during the course of gene editing leading to incomplete reversion to wild type at the epigenetic level, which we speculate was not the case for IsoHD1. Additionally, this observation could also be attributable to a reduced CSF-1 sensitivity in HD2 Mut iMacs as compared to HD1 Mut iMacs and we speculate that this would potentially give rise to the generation

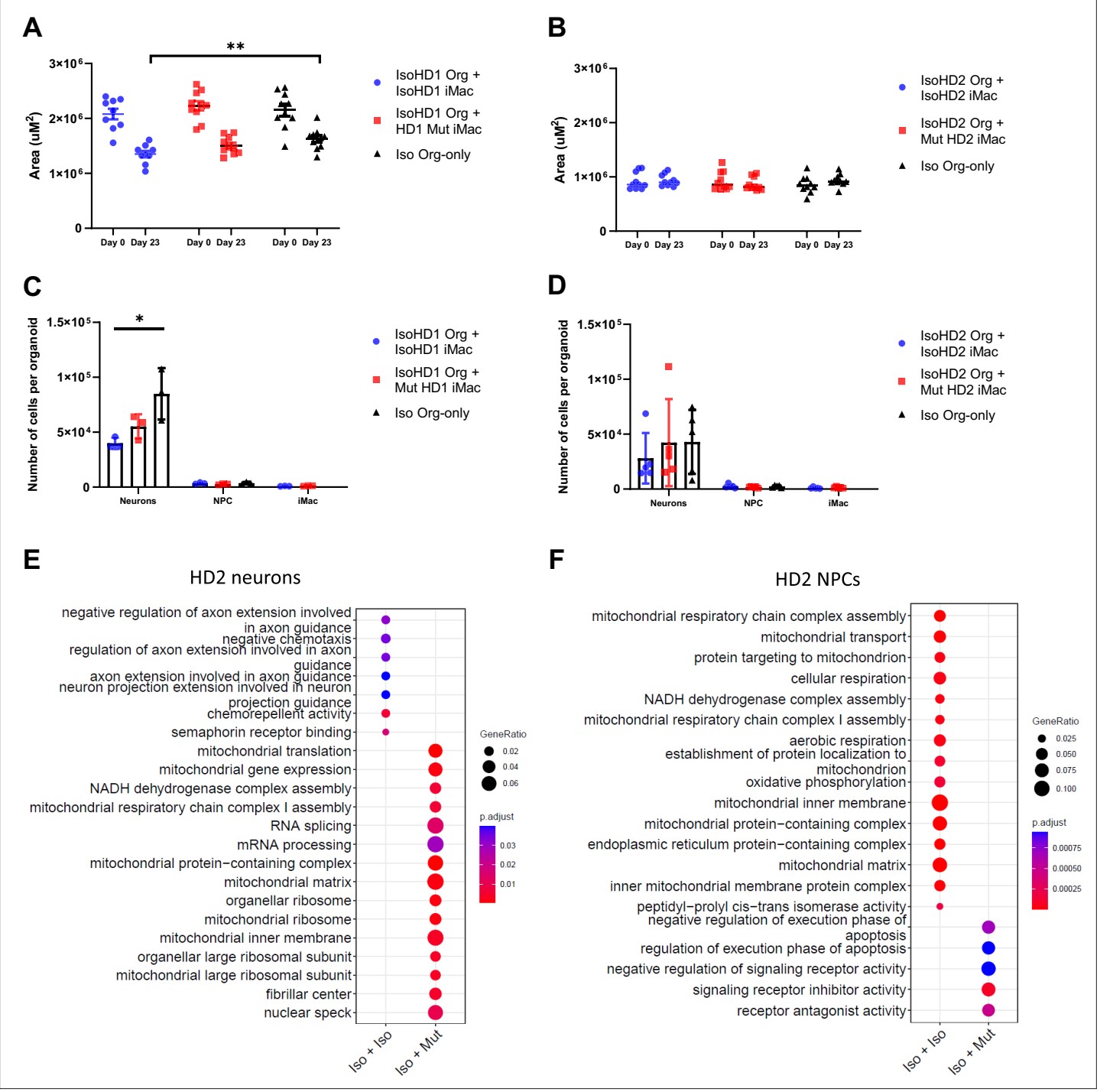

**Figure 6.** Co-culture results in impaired regulation of neurons within organoids. Size of (**A**) HD1 and (**B**) HD2 organoids measured before and after co-culture with mutant and isogenic iMacs in both HD1 and HD2 variants. The effects of co-culture on neuronal populations in (**C**) HD1 and (**D**) HD2 organoids after 23 days. Transcriptomic analysis of co-cultured (**E**) neurons and (**F**) NPCs from HD2 organoids after 23 days. Data are presented as mean ± SEM. Statistical significance was assigned as: p < 0.05 was considered significant: *p < 0.05; **p < 0.01.

and subsequent maintenance of immature iMacs that lacks maturation even after terminally differentiated into tissue-resident macrophages.

Functionally, while phagocytosis of *E. coli* beads was similar across lines, the baseline cytokine production as well as after phagocytosis of apoptotic neuronal cell line was different with an increase of IL-1β secretion in both mutant HD1 and HD2, compared to isogenic HD1 and HD2 iMacs. In the

case of phagocytosis of apoptotic cells, this was not due to a difference in phagocytic activity as all cells were able to phagocytose apoptotic cells at a similar level. Importantly, higher levels of baseline pro- and anti-inflammatory cytokine production in the control group without apoptotic cells were observed within HD1 mutant, and both HD1 and HD2 mutant iMacs, respectively. This suggests that mutant iMacs are innately in a more reactive or activated state when compared to isogenic iMacs and could reflect the reactive phenotype of HDLS-associated microglia and macrophage identified in vivo (*Kempthorne et al., 2020*; *Tada et al., 2016*). Conversely, isogenic iMacs seems to adopt less of the reactive phenotype with the reverted mutation as identified with an overall lesser cytokine production level. Importantly, such reactive phenotype was also highlighted by our transcriptomic and metabolic profiling with upregulated pathways involving pro-inflammatory cytokines and preferential utilization of the glycolytic pathways in the mutant groups, a feature of macrophage activation (*Liu et al., 2021*; *Viola et al., 2019*), respectively.

We subsequently incorporated donor-specific and their isogenic controls into a co-culture system using 3D forebrain organoids and iPSC-derived macrophages to better study HDLS. Isogenic and mutant iMacs co-cultured separately with their respective isogenic forebrain organoid differentiate into microglia-like cells with 23 days of co-culture as shown by the expression of the microglial marker IBA-1, their ramified morphology as well as their transcriptomic profiles with upregulation of genes associated with biological pathways that are typical of microglia functions such as neurodevelopment, neurodifferentiation, neurogenesis, and brain development.

Subsequently, we analyzed the size of the organoids before and after co-culture as we recently showed that addition of iMac modulated NPC differentiation, limiting their proliferation and promoting axonogenesis, resulting in a reduction of co-culture organoid size (*Park et al., 2023*). Here, while we noticed a significant decreased in the size of isogenic HD1 forebrains co-cultured with isogenic HD1 iMac compared to isogenic HD1 organoids, we did not measure any significant change in organoid size in the isogenic HD1 organoids co-cultured with mutant HD1 iMacs. This suggested that the cross-talk between iMicro and NPC is dysregulated in the context of the HD1 mutation. Since we showed that such size reduction is mediated by cholesterol transfer from iMicro to NPC, further work will be required to characterize cholesterol metabolism of HD iMac.

While we did not observe major changes in cell type composition in these co-cultures, transcriptomic analysis of neuronal and NPC population after co-culture showed an increase in mitochondrial respiratory pathways associated genes in the mutant HD2 iMac co-culture neuron and isogenic HD2 iMac co-culture NPCs population, respectively. This suggests increased metabolic rates in these respective populations that are contributed to by the different variants of iMac during co-culture, and show that mutated iMicro can change the states of non-mutated isogenic organoid cells. A parallel process in vivo would likely have profound long-term consequences for brain development and homeostasis.

Altogether, we propose that CSF-1R mutation in HDLS may cause subtle yet impactful disruptions to microglia homeostasis due to abnormal metabolic functions in the developing fetal brain, that subsequently compound and result in aberrant microglia functions during adulthood. This could also be mediated by upregulated pro-inflammatory cytokine production such as IL-1β at a basal level or upon clearing of cellular debris during development and adulthood homeostasis. IL-1β is of particular interest as numerous studies reported the role of IL-1β in neuroinflammation and neurodegeneration especially within the field of Alzheimer's disease: for example, postmortem brain sample analysis revealed elevated levels of Il-1β production, including around amyloid beta plaques; all suggestive of pathology-associated inflammation (*Cacabelos et al., 1994*; *Shaftel et al., 2008*; *Italiani et al., 2018*). Hence, we speculate that in CSF-1R$^{+/-}$ mutant microglia, which already have a predisposition toward increased secretion of IL-1β, are constantly being activated by IL-1β, either autocrine or paracrine, creating a dysfunctional neuro-environment that contributes to HDLS pathology. As such, experimental models looking to block or reduce IL-1β levels in the brain using the human mAb, canakinumab, targeting the pro-inflammatory cytokine IL-1β (*Ridker et al., 2011*; *Yuan et al., 2022*), during both health and disease could be a potential target for therapy and may help to ameliorate HDLS pathology.

While no model perfectly recapitulates the full complexity of a disease, the application of iPSC-derived organoid co-culture approaches to primary patient-derived cells offers a valuable way to study the disease mechanisms of this rare pathology using human cells. Future studies employing

microglia-sufficient patient-derived brain organoids have the potential to shed much-needed light into the pathogenesis of HDLS and other neurodegenerative diseases.

## Materials and methods

### Preparation of human dermal fibroblasts from donor biopsies

Samples from patients with HDLS were collected via skin punch and placed in biopsy medium [Gibco, DMEM Glutamax supplemented with 1% 100× Antibiotics/Antimycotics (Gibco, AB/AM) and 10% fetal bovine serum (Gibco, FBS)] for transport to the laboratory. Immediately on arrival, biopsies were placed in 75% ethanol for 30 s then washed three times with DMEM Glutamax supplemented with 1% AB/AM; attached adipose tissue was removed using sterile scissors before biopsies were cut into small 1 mm$^2$ cubes and placed dermis-side down onto a 6-well plate then left to dry slightly for 20 min to facilitate adhesion. Biopsy culture medium was then added and plates incubated overnight at 37°C, 5% carbon dioxide in air; 48 hr later, the medium was changed then biopsies cultured for a further 10–14 days during which fibroblast outgrowth was observed. At around 90% confluency, fibroblasts were detached from the plates using 0.05% Trypsin/EDTA and passaged. Biopsies were re-plated up to three times for further fibroblast propagation.

### Reprogramming of human dermal fibroblasts into induced pluripotent stem cells

Individual donor's fibroblasts were reprogrammed into iPSCs using the Epi5 Episomal reprogramming kit (Invitrogen), and the Human dermal fibroblast nucleofector kit (Lonza) on a Lonza Nucleofector 2B device. Briefly, fibroblast cultures at approximately 90% confluency were detached from their plates using 0.025% Trypsin/EDTA then collected by centrifugation at 350 × *g* for 5 min at room temperature. Cell pellets were resuspended in 100 µl of nucleofector solution at room temperature then counted: 200,000 cells were used per transfection with 1 µl of each of the episomal vectors, using program U-023. Transfected cells were subsequently plated onto Matrigel-coated (Stemcell) 6-well plates with 2 ml of pre-warmed fibroblast medium (DMEM supplemented with 10% FBS and 1% AB/AM) and incubated for 2 days. The medium was then replaced with 2 ml of TeSR-E7 medium, which was changed every 2 days for 25–35 days until early iPSC clones emerged.

### Isolating iPSC colonies

Between days 25 and 35, colonies resembling human embryonic stem cell colonies emerged, were collected using a sterile needle under a stereomicroscope, mechanically fragmented, and added to 24-well plates pre-coated with Matrigel and containing 500 µl of mTeSR-1. The colonies were cultured for 24 hr then underwent medium change with fresh mTeSR-1 for 7–10 days until 80% confluent.

### Culture and maintenance of iPSC cell lines

To passage, 500 µl of ReleSR were added into each well and incubated in 37°C for 3 min before tapping the plate firmly on all sides to dislodge the iPSC colonies. Subsequently, the iPSC fragments were transferred into 12-well plates and again cultured until approximately 80% confluent, before passaging into 6-well plates, all while maintaining in mTeSR-1 with daily medium change. Mature iPSC colonies were identified by their characteristic phenotype: clear distinct borders, high nuclei-to-cytoplasm ratio, and raised, refractive, three-dimensional-like colonies.

### Characterization of induced pluripotent stem cell colonies

Characterization of iPSC colonies was achieved using the PSC 4-Marker Immunocytochemistry Kit (Gibco). Briefly, spent mTeSR-1 medium was aspirated from the cells before adding 200 µl of fixative solution per well and incubating for 15 min at room temperature, which was then replaced by 200 µl of permeabilization solution for 15 min, then by 200 µl of blocking solution for 30 min. Subsequently, 2 µl of antibodies recognizing SSEA4 and OCT4 were added directly to the blocking solution and incubated for 3 hr before washing three times with wash buffer (1× DPBS). Secondary antibodies were added at 0.8 µl per well and incubated for 1 hr at room temperature before washing three more times. During the last wash, one drop of NucBlue Fixed Cell Stain (DAPI) was added into the wash buffer

and incubated for 5 min. Cells were either imaged immediately or stored at 4°C in the dark, for up to 1 month before analysis.

## Karyotyping iPSC colonies

Mature iPSC colonies (HD1, IsoHD1, HD2, IsoHD2) were pre-treated with 0.1 µg/ml of colcemid for 3 hr in the incubator before dissociation into single cells for imaging. Subsequently, cells were fixed with 4% paraformaldehyde and sent for karyotype analysis at Zhen He Biotech, Shanghai.

## Microscopy

Cell morphology observations were regularly performed using an Olympus CKX41 inverted microscope. Bright field and phase contrast images were taken using a Nikon Eclipse TS2 inverted microscope equipped with a Digital Sight camera. Images were processed using the proprietary software ImageView.

## Mycoplasma analysis

The absence of mycoplasma infection was assessed by MycoStrip Mycoplasma Detection Kit (InvivoGen, rep-mys-10), according to the manufacturer's instructions (data not shown, available on request).

## Differentiation of iPSC into primitive-like macrophages (iMacs)

Patient-derived iMacs were generated as previously described (*Takata et al., 2017*). Briefly, iPSCs were cultured in Stempro34 medium, supplemented with L-glutamine (100×), ascorbic acid (5 mg/ml), 1-Thioglycerol (26 µl/2 ml), Transferrin (200×) and Pen/Strep (1%), with the addition of CHIR99021 (2 µM), VEGF (50 ng/ml), and BMP4 (5 ng/ml) from days 0 to 2 to induce mesodermal lineage commitment; hemangioblast-like cell formation was then induced by culture in the presence of BMP4 (5 ng/ml), VEGF (50 ng/ml), and bFGF (20 ng/ml) on day 2 and only VEGF (15 ng/ml) and bFGF (5 ng/ml) on day 4. From days 6 to 10, hemangioblasts were further committed to the generation of hematopoietic cells by the addition of DKK-1 (30 ng/ml), VEGF (10 ng/ml), bFGF (10 ng/ml), IL-6 (10 ng/ml), IL-3 (20 ng/ml), and SCF (50 ng/mL). To further promote the maturation of hematopoietic cells and CSF-1R expression, on days 12 and 14, medium was supplemented with SCF (50 ng/ml), bFGF (10 ng/ml), IL-6 (10 ng/ml), and IL-3 (20 ng/ml). From days 16 to 31, the medium was switched to SF-Diff consisting of IMDM (75%) and F12 (25%) with the addition of N2 (100×), B27 (50×), 0.05% BSA, 1% Pen/Strep, supplemented with 50 ng/ml of CSF-1 to terminally differentiate the primitive cells to iMacs. Full medium changes were carried out every 2 days from days 0 to 14 and every 3 days from days 16 to 31. Free-floating cells were observed from as early as day 6 of culture and were collected and re-seeded back into individual wells during medium changes. Additionally, iMacs were initially cultured in a hypoxia incubator (5% $O_2$, 5% $CO_2$) from days 0 to 8 before moving into a normoxic incubator until day 31, when the cells were used for experiments. iMac identity was confirmed by flow cytometry with labeling for the macrophage markers CD45 and CD14.

## Generation of forebrain organoids

Forebrain organoid generation was adapted from *Qian et al., 2018*, which described a guided differentiation method that requires various cytokines and growth factors to be added at different differentiation stages, as follows.

## Stage 1 (days 0–4: embryoid body formation)

iPSC on 6-well plate at approximately 80% confluency were dissociated into single cells using Accutase (Stemcell) before counting and seeding at 50,000 cells per well into an ultra-low attachment (ULA) 96-well plate with 150 µl of forebrain first medium (F1M: DMEM/F12, 20% KOSR, 1× GlutaMax, 1× MEM-NEAA, 1× 2-Mercaptoethanol, Pen/strep, Dorsomorphine 2 µM, A-83 2 µM) and 20 µM of ROCK Inhibitor (Stemcell). Plates were incubated at 37°C and 5% $CO_2$ for 24 hr (day –1). On day 0, medium was refreshed with the addition of 2 µM Dorsomorphin (Sigma) and 2 µM A83-01 (Sigma). Half medium changes were then performed on days 2 and 4 with the exclusion of ROCK inhibitor: small embryoid bodies (EB) were observed at this point.

## Stage 2 (days 5–13: neuro-ectodermal induction)

On day 5, one EB was transferred into each well of a ULA 24-well plate with 500 µl of Forebrain second medium (F2M: DMEM/F12, 1× N2 supplement, 1× GlutaMax, 1× MEM-NEAA, Pen/Strep, CHIR-99021 1 µM, SB-431542 1 µM). On day 7, healthy EB displaying a smooth and round surface were embedded individually into 20 µl of Matrigel, according to the method of *Lancaster et al., 2013*. Once the Matrigel had fully polymerized, eight embedded EBs were transferred into a 10-cm Petri dish with 10 ml of F2M before incubating at 37°C with half medium changes every 2 days until day 13. At this point, cluster(s) of neuroepithelium buds were present in individual organoids, evident as neural tube-like structure without extending cell processes.

## Stage 3 (days 14–31: maturation)

On day 14, Matrigel-embedded organoids were collected and replated, with the attached Matrigel removed by repeated pipetting, before being transferred into a clean 10-cm Petri dish with 8 ml of Forebrain third medium (F3M: DMEM/F12, 1× N2 supplement, 1× B27 supplement, 1× GlutaMax, 1× MEM-NEAA, 1× 2-Mercaptoethanol, Pen/Strep, Insulin 2.5 µg/ml). Dishes were incubated at 37°C on an orbital shaker at 80 rpm, with half F3M changes every 3 days until organoids were used for analysis/experiments.

## Generation of isogenic clones using CRISPR/CAS9

sgRNAs were designed using the CRISPR design tool (http://crispor.tefor.net/) and selected based on highest efficiencies as well as off-target scores and synthesized by GENESCRIPT with scaffold. ssDNA template was designed with 40 bp of homologous nucleotides flanking both arms of the point of mutation, inclusive of silent mutations on, and adjacent to, the PAM region, and synthesized by Sangor Shanghai. Genome editing via homology-directed repair from the CRISPR/Cas9 system in combination with a ssDNA was used to guide the single nucleotide correction of the HDLS point mutation, according to donor's Sanger sequencing reports on Exon 15 (Donor #2) and 18 (Donor #1) of CSF-1R by homologous recombination.

The ribonucleoprotein (RNP) complex consisting of Cas9 protein and sgRNA was assembled using the Trucut V2 (Invitrogen) and Amaxa Human Stell Cell Nucleofector Starter Kit (VPH-5002). The RNP complex was subsequently transfected into dissociated iPSCs by electroporation with Lonza Nucleofector 2B using the Amaxa Human Stem Cell Nucleofector Starter Kit (Lonza) according to the manufacturer's manual. After electroporation, the cells were immediately plated onto Matrigel-coated 6-well plates containing mTeSR1 medium supplemented with 10 µM ROCK inhibitor (Stemcell) and CloneR (Stemcell) at various seeding densities to facilitate the harvesting of clones. After 48 hr, medium was refreshed with the removal of ROCK inhibitor; subsequently, medium was refreshed daily with the removal of both ROCK inhibitor and CloneR until clones were mature enough for manual picking. Individual clones were manually harvested and plated onto Matrigel-coated 24-well plates with mTesr-1 medium and 10 µM of ROCK inhibitor.

Clones were screened for SNP editing using the ICE SYNTHEGO tool (https://ice.synthego.com) and off-target events screened for using CRISP-ID software (http://crispid.gbiomed.kuleuven.be).

## Co-culture of forebrain organoids and iMacs to generate iMicroglia

Co-culturing iMacs (primitive macrophages) with organoids enables the iMacs to be in a neurogenic niche environment where they would subsequently differentiate into iPSC-derived microglia (iMicro). Thirty thousand day 31 iMacs (mutant or Isogenic controls) were resuspended in F3M before adding onto their isogenic organoid in each well of an ULA 96-well plate. Co-cultures were incubated for 7 days with daily half F3M change supplemented with CSF-1 (100 ng/ml). On day 8, each organoid was transferred to an individual well of a ULA 24-well plate and supplemented with 1 ml of F3M and CSF-1 (100 ng/ml); half F3M change was performed every 3 days until analysis.

## Clearing of forebrain organoids for 3D imaging

Organoids were first fixed in 4% paraformaldehyde for 30 min at room temperature followed by washing twice in PBS then incubating in PBS overnight at 4°C. The next day, PBS was replaced with a permeabilization buffer (2% Triton X-100 in PBS solution) and organoids were incubated on an orbital shaker at room temperature for 72 hr. At the end of permeabilization, organoids were incubated in

blocking buffer (5% normal donkey serum, 5% normal goat serum, 1% FBS, 1% Triton X-100 in PBS solution) at 4°C overnight. Organoids were then incubated with primary antibodies at 1:200 in antibody dilution buffer (1% bovine serum albumin, 0.2% Triton X-100 in PBS solution) at 4°C for 72 hr. Primary antibodies specific for the following markers were used: NESTIN (MERCK), SOX2 (Abcam), and IBA-1 (Abcam). Organoids were washed in washing buffer (1× PBS solution) on an orbital shaker at room temperature for 1 hr before being kept in 1× PBS solution at 4°C overnight. The organoids were subsequently incubated with the following secondary antibodies at 1:200 in antibody dilution buffer, at 4°C for 48 hr: goat anti-mouse Alexa Fluor 488 (Biolegend), goat anti-rabbit Alexa Fluor 594 (Abcam), and donkey anti-goat Alexa Fluor 647 (Abcam). The organoids were then washed with wash buffer for 1 hr on an orbital shaker at room temperature then kept in PBS at 4°C overnight before counterstaining with the nuclear marker DAPI on an orbital shaker at room temperature for 2 hr. Organoids were then washed twice in washing buffer on an orbital shaker at room temperature for 1 hr, then again kept overnight at 4°C in 1× PBS. The following day, organoids were transferred individually to confocal-imaging-compatible glass-bottomed containers and 20 µl of RapiClear (SUNJin Lab) solution was added onto each organoid and left until they were cleared (usually 24 hr depending on the size of the organoid). Whole organoids were imaged using a FV3000 confocal microscope with 20× objective lens (Olympus, Japan) and analyzed using iMaris software (BITPLANE).

## Measurement of area of organoid

Area of organoids was measured using ImageJ software. Briefly, the 'freehand selection' tool was used to trace the circumference of each individual organoid. The average of three measurements was used as the final readout.

## Dissociation of forebrain organoids

Organoids were collected individually using 200 µl pipette with a cut pipette tip and incubated in 500 µl of Accutase at 37°C with gentle pipetting after 10 min, then allowed to fully dissociate for another 5–15 min at 37°C as needed. After again pipetting, cells were suspended in 1 ml of MACS buffer and collected by centrifugation at 1350 rpm 350 × $g$ at RT for 5 min. Cells were then used in experiments.

## Flow cytometry

Cells were prepared for labeling by first washing in FACS buffer, then resuspending cells in 70 µl of Fc-block (1:200) and incubating for 20 min at 4°C. Subsequently, the primary antibody cocktails were added into the cells at a 1:200 dilution before being vortexed and incubated for 30 min at 4°C. For samples that required secondary antibody labeling, after incubation with primary antibodies the cells were washed with FACS buffer before secondary antibodies (1:500) were added and incubated for 25 min at 4°C. After a further wash in FACS buffer, DAPI was added then cells were analyzed using the BD Symphony X-50.

## Cell sorting of co-cultured organoids

Cells were isolated into sorting solution (SMART-Seq HT kit, 10× lysis buffer, RNase inhibitor, 3' SMART-Seq CDS Primer II A, nuclease-free water) using the BD Aria III cell sorter. After sorting, cell-containing tubes were briefly spun down and immediately placed in liquid nitrogen for snap freezing. Lysates were stored at –80°C.

## RNA extraction of samples

RNA extraction was performed using Trizol according to the manufacturer's instructions. Briefly, 500 µl of Trizol was added to each sample before the addition of 100 µl of chlorofoam to each sample and mixed thrice (15 s per succession) before incubating at room temperature for 5 min. Samples were then centrifuged at 20,000 × $g$, 20 min, 4°C before the organic solvent was transferred into new eppendorf tubes. One µl of GlycoBlue co-precipitate (Invitrogen) was added to each sample and mixed well before the addition of 250 µl of isoproponol. Samples were mixed well before incubating at –80°C for at least 1 hr. Thawed samples were subsequently centrifuged at 12,000 × $g$, 4°C for 15 min before decanting supernatant, washing again with ice-cold 75% ethanol

and centrifuging at 7500 × $g$, 4°C for 5 min. Samples were left to air dry for approximately 5 min after decanting of supernatant, before the RNA pellet was resuspended in 10 µl of DPEC water and stored at –80°C.

## cDNA library construction for RNA-sequencing

RNA samples from the previous step were first converted to complementary DNA (cDNA) using the Takara HT kit according to the manufacturer's instruction. Briefly, 5 µl of purified total RNA was added to 1 µl of 10× reaction buffer consisting of 10× lysis buffer and RNase inhibitor; 1 µl of 3' SMART-seq primer II A was added and mixed well by vortexing before incubating the samples at 72°C for 3 min. Samples were then immediately placed on ice for 2 min. A one-step master mix consisting of 0.7 µl of nuclease-free water, 8 µl of one-step buffer, 1 µl of SMART-Seq HT oligonucletide, 0.5 µl RNase inhibitor, 0.3 µl SeqAmp DNA polymerase, and 2 µl of SMARTscribe reverse transcriptase was subsequently added to each sample and mixed well. The following program was used to run the PCR protocol:

| 42°C | | 90 min |
|---|---|---|
| 95°C | | 1 min |
| | 98°C | 10 s |
| | 65°C | 30 s |
| 18 cycles | 68°C | 3 min |
| 72°C | | 10 min |
| 4°C | | Forever |

The resulting cDNA product was purified using the Vazyme VAHTS DNA Clean beads. Briefly, 20 µl of beads were added to each cDNA product, mixed well and incubated for 5 min at RT before placing samples on a magnectic stand for 2 min. The supernatant was discarded and cDNA was washed twice with cold 80% ethanol before being air-dried for 3 min. Purified cDNA was subsequently eluted from the magnetic beads with 15 µl of DPEC water. The concentration of cDNA was measured with the Qubit dsDNA HS Assay Kit (Invitrogen) using the Qubit 4 fluorometer.

cDNA library construction was performed using the Vazyme TruePrep DNA library prep kit V2 for illumina according to the manufacturer's instructions. Briefly, the purified cDNA obtained from the previous step was diluted to a final concentration of 1 ng/µl with sterile water. Next, 7.5 µl of Mix A (2 µl 5× TTBL, 2.5 µl TTE Mix V5, 3 µl DPEC water) was added to 2.5 µl of diluted cDNA and run using the Nextera-1 program at 55°C for 10 min.

TSS was immediately added to each sample and left to incubate at RT for 5 min before 7.5 µl of Mix B (2 µl DPEC water, 5 µl 5× TAB, 0.5 µl TAE) was added to each sample. 2.5 µl of N5 and N7 primer was added and mixed well before running on the Nextera-2 program:

| 72°C | | 3 min |
|---|---|---|
| 98°C | | 3 min |
| | 98°C | 15 s |
| | 60°C | 30 s |
| 13 cycles | 72°C | 3 min |
| 72°C | | 5 min |
| 4°C | | Forever |

Subsequently, samples were purified using the vazyme beads and cDNA was eluted with 20 µl of DPEC water before measuring the concentration.

For pooling of cDNA libraries, 40 ng of each sample was added to a final volume of 40 µl, adjusted with DPEC water, and incubated with 22.6 µl of Vazyme clean beads at RT for 5 min before magnectic separation. The entire volumn of supernatant was transferred to a fresh PCR tube to which was added 6.4 µl of Vazyme clean beads. The contents were mixed well and incubated at RT for 5 min before separation: subsequently, the supernatant was discarded and bead-bound-cDNA was washed twice

with 80% ethanol before being air dried for 3 min. Purified pooled cDNA samples were eluted with 20 µl of DPEC water and tested for concentration.

## Bulk RNA analysis

The clean paired-ends reads were aligned to the GRCh38.96 human genome reference using kallisto (version 0.46.1) with parameters '–bootstrap-samples=100'. The transcript-level estimated counts belonging to the same gene were then aggregated into the matrix of gene-level counts (TPM) implemented in the R package tximport (version 1.14.2).

Pairwise comparison across RNA-seq data was performed to get differentially expressed genes between each two types of cells, using the linear model and the empirical Bayes method implemented in R limma package (version 3.42.2), with significance thresholds for p-value <0.05 and $\log_2$(fold change) $\geq 0.5$ (*Figure 3*) or 1.5 (*Figure 5*). The differentially expressed genes identified with top $\log_2$(fold change) were selected to perform the PCA analysis in R using prcomp function. Additionally, the clusterProfiler package (version 4.10.0) was employed for pathway enrichment analysis via the compareCluster function and 'enrichGO' method, targeting 'ALL' GO ontologies.

## Metabolic function using Seahorse analysis

The bioenergetic profile of iMacs was assessed by determining the OCR and the ECAR using an XF-96 Flux Analyzer (Seahorse Bioscience). Sensor cartridges (Agilent Technologies) were hydrated in XF Calibrant (Agilent Technologies) at 37°C overnight, devoid of carbon dioxide, following the manufacturer's instructions. Cells were resuspended in either OCR XF base medium supplemented with 10 mM glucose, 2 mM L-glutamine, and 1 mM sodium pyruvate, or ECR XF base medium supplemented with 2 mM L-glutamine, before being seeded at 40,000 cells/50 µl/well into XF96 culture plates. Cells were pelleted by centrifugation at 200 × $g$ for 1 min then resuspended in either 125 µl of OCR or ECR medium per well and incubated at 37°C in a $CO_2$-free incubator for 45 min before loading into the Seahorse analyzer. Cells were sequentially treated with 25 µl injections of specific bioenergtic modulators that were added to the wells prior to loading, to test for different paramters of mitochondrial and glycolytic functions. ECR analysis was probed with 10 mM glucose to stimulate glycolysis, followed by the addition of 2 µM oligomycin to block ATP synthase and 50 mM 2-deoxy-glucose to shut down the function of glycolysis. Conversely, OCR analysis was probed with 2 µM oligomycin, 1 µM FCCP (carbonyl cyanide-4-(trifluoromethoxy) phenylhydrazone) to stimulate maximal mitochondrial oxygen consumption and 0.5 µM rotenone plus antimycin A to asses OXPHOX parameters from the OCR levels. Mitochondrial and glycolytic parameters were calculated as recommended by the instrument manufacturer (Agilent Technologies).

## MTT test

On the day before testing, 25,000 iMacs were seeded into each well of a 96-well flat-bottom plate. The next morning, 20 µl of MTT solvent (10% of total volume) was added to each well and incubated at 37°C for 3.5 hr before being decanted. Next, 200 µl of MTT formazan solvent was added to each well before being placed onto a shaker for 15 min in the dark, then incubated RT for an additional 30 min in the dark. Measurements were read by a spectrophotometer at OD570 and OD690.

## Cytospin assay

iMacs were first trypsinzed using TryPLE express before being counted using the Countess II. Approximately 50,000 cells were resuspended in 150 µl of PBS before being transferred to a cytocentrifuge cytofunnel paired with a glass slide, and centrifuged for 5 min at 800 rpm 200 × $g$, air dried for 30 min before sealing with mounting medium. Images were acquired with an Olympus BX53 light microscope equipped with a 100× oil immersion objective lens.

## Giemsa staining

Adequately air-dried slides were placed in a staining tray and flooded with 1 ml per slide of Solution A of the Wright-Giemsa staining kit (BASO) for 1 min. Subsequently, 500 µl of Solution B was added to each slide and agitated for 5 min. Slides were then rinsed in distilled water for 5 min before being air-dried for 30 min then sealed in synthesis resin. Images were acquired with the Olympus BX53 light microscope paired with a 100× oil immersion objective lens.

## Neuronal cell line SH-5YSY apoptosis induction

SH-SY5Y (ATCC, CRL-2266) cells were kindly provided by the lab of Xu Tien le from the Shanghai Institute of Immunology. SH-SY5Y cell monolayers in sterile PBS were exposed to UV radiation for 30 min before collection of cells into serum-free medium (DMEM + 1% P/S) and incubation at 37°C overnight to induce apoptosis. The following day, cells were centrifuged at $350 \times g$, RT for 3 min before staining with Annexin V (BD Pharmingen PE Annexin V apoptosis detection kit) for 20 min at RT in the dark, before being analyzed by flow cytometry.

## Phagocytic assay with conjugated *E. coli* beads

First, pHrodo Red *E. coli*-conjugated bioparticles (Invitrogen) were reconstituted to a working concentration of 1 mg/ml with sterile water before adding to iMacs in complete medium at 1:10 final concentration. After incubation at 37°C for 45 min, cells were washed twice with PBS before collection for flow cytometry analysis.

## RNA extraction and real-time quantitative PCR

Total RNA was extracted from each group with TRIzol Reagent (Invitrogen). The cDNA was obtained by using the PrimeScript RT Reagent Perfect Real Time kit (Takara, Japan). Briefly, 500 ng of RNA for each sample was amplified as a template, in 10 µl of reaction mixture (2 µl 5× PrimeScript buffer, 0.5 µl PrimeScript RT Enzyme Mix I, 0.5 µl Oligo dT Primer, 0.5 µl random 6-mers, RNA product, RNase-free water). This mixture was then incubated at 37°C for 15 min and subsequently at 85°C for 5 sec to obtain cDNA. Quantitative PCR (qPCR) was performed in a ViiA 7 Real-time PCR system (Applied Biosystems) with ChamQ SYBR Color qPCR Master Mix Low ROX Premixed (Vazyme).

## Statistical analysis

Statistical tests were performed using GraphPad Prism 8.0.1 (GraphPad Software, La Jolla, CA, USA; https://www.graphpad.com/). All experiments were performed in triplicate and data analysis was performed using unpaired Student's $t$-tests. All data represent the mean ± SEM. Statistical significance was assigned as: $p < 0.05$ was considered significant: *$p < 0.05$; **$p < 0.01$; ***$p < 0.001$. $p > 0.05$ was considered nonsignificant (n.s.).

## Additional information

### Competing interests

Florent Ginhoux: Reviewing editor, eLife. The other authors declare that no competing interests exist.

### Funding

No external funding was received for this work.

### Author contributions

Wei Jie Wong, Data curation, Formal analysis, Investigation, Writing – original draft, Project administration, Writing – review and editing; Yi Wen Zhu, Hai Ting Wang, Jia Wen Qian, Ziyi Li, Song Li, Wei Guo, Shuang Yan Zhang, Formal analysis; Zhao Yuan Liu, Formal analysis, Supervision, Project administration; Bing Su, Supervision, Project administration; Fang Ping He, Kang Wang, Resources, Project administration; Florent Ginhoux, Conceptualization, Resources, Writing – original draft, Project administration, Writing – review and editing

### Author ORCIDs

Wei Jie Wong ⓘ https://orcid.org/0009-0003-5342-7948
Bing Su ⓘ https://orcid.org/0000-0003-0871-7666
Kang Wang ⓘ https://orcid.org/0000-0002-3664-0638
Florent Ginhoux ⓘ https://orcid.org/0000-0002-2857-7755

### Ethics

This study was approved by the Medical Ethics Committee of the First Affiliated Hospital, College of Medicine, Zhejiang University (Approval Number: 2016-7). The informed consent and consent to publish were obtained from all donors.

Reviewer #1 (Public Review): https://doi.org/10.7554/eLife.96693.3.sa1
Reviewer #2 (Public Review): https://doi.org/10.7554/eLife.96693.3.sa2
Author response https://doi.org/10.7554/eLife.96693.3.sa3

## Additional files

### Supplementary files

MDAR checklist

Supplementary file 1. Supplementary tables.

### Data availability

Sequencing data have been deposited in GEO under accession code GSE271810.

The following dataset was generated:

| Author(s) | Year | Dataset title | Dataset URL | Database and Identifier |
|-----------|------|---------------|-------------|--------------------------|
| Wong W, Li Z, Ginhoux F | 2024 | Modeling Hereditary Diffuse Leukoencephalopathy with Axonal Spheroids using microglia-sufficient brain organoids | https://www.ncbi.nlm.nih.gov/geo/query/acc.cgi?acc=GSE271810 | NCBI Gene Expression Omnibus, GSE271810 |

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
