## [Editor Report · eLife assessment]

This study presents **valuable** findings on the mechanisms underlying a rare brain disease using an organoid system. In this revised version, there are remaining reviewers' comments that are not yet addressed and as such, while the data presented are **solid**, the evidence supporting some of the claims is deemed incomplete. The work will be of interest to neuroscientists and clinicians aiming to understand and combat similar neurodegenerative disorders.

---

## [Referee Report · Reviewer #1 (Public Review)]

Here, using an organoid system, Wong et al aimed to establish new models of hereditary diffuse leukoencephalopathy with axonal spheroids (HDLS), with which they wanted to understand how CSF1R-mutaions affect the phenotypes of microglia/macrophages. They found metabolic changes in microglia/macrophages with mutations, which were associated with a proinflammatory phenotype. In general, the authors tackle important issues and provide valuable tools to investigate the underlying mechanisms for HDLS.

Strength:

The authors establish two HDLS patient-derived iPS cells with their isogeneic controls and provide possible mechanistic insights into the disease mechanisms.

Weakness:

It is unclear how nicely the organoid system in this study can recapitulate the condition in patients with HDLS (e.g. reduced microglia density, downregulated expression of P2YR12, pathological alterations).

The authors generated two different models with distinct mutations that produce different readouts in CSF1R-mediated cellular responses. It is unclear if the different outcomes between HD1 and HD2 are generated simply through different mutations or due to different differentiation efficiency from iMacs.

Suggestions:

(1) This paper would benefit from additional histological analyses to characterize iMac & iMicro at least histologically, which would be helpful for readers to know how nicely the organoid system recapitulates the condition in patients with HDLS.

(2) In addition, in Fig.5E-J the authors could highlight microglia core genes that would be upregulated if iMacs are successfully differentiated into iMicro.

(3) Since there are no direct evidence to confirm that "microglial dysregulation and IL1b signalling contribute to the degenerative neuro-environment in HDLS", the authors should tone down their argument and rephrase the Abstract.

---

## [Referee Report · Reviewer #2 (Public Review)]

Summary:

This paper investigates a rare and severe brain disease called Hereditary Diffuse Leukoencephalopathy with Axonal Spheroids (HDLS). The authors aimed to understand how mutations in the gene CSF-1R affect microglia, the resident immune cells in the brain, and which alterations and factors lead to the specific pathophysiology. To model the human brain with the pathophysiology of HDLS, they used the human-specific model system of induced pluripotent stem cell (iPSC)-derived forebrain organoids with integrated iPSC-derived microglia (iMicro) from patients with the HDLS-causing mutation and an isogenic cell line with the corrected genome. They found that iPSC-derived macrophages (iMac) with HDLS mutations showed changes in their response, including increased inflammation and altered metabolism. Additionally, they studied these iMacs in forebrain organoids, where they differentiate into iMicro, and showed transcriptional differences in isolated iMicro when carrying the HDLS mutation. In addition, the authors described the influence of the mutation within iMicro on the transcriptional level of neurons and neural progenitor cells (NPCs) in the organoid. They observed that the one mutation showed implications for impaired development of neurons, possibly contributing to the progression of the disease. Overall, this study provides valuable insights into the mechanisms underlying HDLS and emphasizes the importance of studying diseases like these with a suitable model system. These findings, while promising, represent only an initial step towards understanding HDLS and similar neurodegenerative diseases, and thus, their direct translation into new treatment options remains uncertain.

Strengths:

The strength of the work lies in the successful reprogramming of two HDLS patient-derived induced pluripotent stem cells (iPSCs) with different mutations, which is crucial for the study of HDLS using human forebrain organoid models. The use of corrected isogenic iPSC lines as controls increases the validity of the mutation-specific observations. In addition, the model effectively mimics HDLS, particularly concerning deficits in the frontal lobe, mirroring observations in the human brain. Obtaining iPSCs from patients with different CSF1R mutations is particularly valuable given the limitations of rodent and zebrafish models when studying adult-onset neurodegenerative diseases. The study also highlights significant metabolic changes associated with the CSF1R mutation, particularly in the HD2 mutant line, which is confirmed by the HD1 line. In addition, the work shows transcriptional upregulation of the proinflammatory cytokine IL-1beta in cells carrying the mutation, particularly when they phagocytose apoptotic cells, providing further insight into disease mechanisms.

Weaknesses:

Most of the points have been addressed in the revision, but some points remain (see below) and are well within the scope of the current manuscript in this reviewer's opinion.

(1) The characterization of iMicros is incomplete, with limited protein-level analysis (e.g. validate RNA-seq data via flow cytometry, ELISA etc.).

(2) Additionally, the claim of microglial-like morphology lacks adequate evidence, as the provided image is insufficient for such an assessment (also the newly provided Supp. Fig. 3C is insufficient and looks rather like background). Show single channels for each staining. Show examples for both cell lines.

(3) RNA-seq experiments are still difficult to read. A combination of data from both lines into one big analysis would be advantageous. E.g. showing overlapping GO terms for both lines. What is common, what is different in both lines?

(4) Statistical test information is missing in the legends.

---

## [Author Response]

The following is the authors’ response to the original reviews.

We thank the reviewers for their feedback on our manuscript. Taking the advice of the reviewers, we have streamlined the text and formatted the figures to conform to the format instructions. We believe that the revised manuscript has been improved.

Point-by-point responses are presented below.

**Reviewer #1:**
(1) There is an over-interpretation regarding the results in Figure 6A. There is no difference between isoHD1 iMac control and HD1 Mut iMac.

We thank the reviewer for his/her feedback on our manuscript. We have since changed the wordings on Page 11, line 294 of the manuscript, to reflect this important point.

**Reviewer #2:**
(2) The authors have not elucidated the significance of the increased CSF1 dosage in Figure 2F, aside from its effect on cell viability, lacking a thorough discussion of this result.

We have incorporated the significance of the results of our CSF1 dosage data with a newly added observation of an upregulated immature myeloid marker and downregulated expression mature macrophage marker within mutant iMac from the respective RNA-seq data (Page 5, line 163); and elaborated further within the Discussion section that this results in the possible generation of immature iMacs even after maturation (Page 14, line 356).

(3) Additionally, while transcriptomic and metabolic alterations related to the mutation were demonstrated in iMac models, similar investigations in iMicros are absent, necessitating further experiments to validate the findings across cell models.

We thank the reviewer for this feedback and feel that this is beyond the scope of this study at current stage, and that we would keep this in consideration to incorporate into subsequent experiments.

(4) The conclusion drawn regarding cytokine levels lacks robust support from the data, particularly considering the varied responses observed in different mutant lines. Further analysis of the secretome (e.g. via ELISA) could provide additional insights.

We thank the reviewer for this feedback and feel that this is beyond the scope of this study at current stage, and that we would keep this in consideration to incorporate into subsequent experiments.

(5) Moreover, the characterization of iMicros is incomplete, with limited protein-level analysis (e.g. validate RNA-seq via flow cytometry).

We thank the reviewer for this feedback and feel that this is beyond the scope of this study at current stage, and that we would keep this in consideration to incorporate into subsequent experiments.

(6) Additionally, the claim of microglial-like morphology lacks adequate evidence, as the provided image is insufficient for such an assessment.

We have added confocal images depicting microglial-like morphology in our co-culture system within Supp Fig 3C.

(7) RNA-seq experiments should be represented better, it is not possible to read the legends or gene names in the figures. Maybe the data sets can be combined into PCAone and one overall analysis, e.g. via WGCNA-like analyses? This would make it easier for the reader to compare the two cell lines side by side.

We have since enhanced the quality of the respective RNA-seq figures with enlarged data points and gene names for better clarity.

(8) Statistical test information is missing.

We are sorry for leaving this out and have added the statistical test information within Page 15 of the methods section.

(9) Finally, inconsistent terminology usage throughout the paper may confuse readers (iMac versus iMicros).

We have streamlined the terminology used within Page 10, line 265 and 267, of the manuscript for better consistency.

(10) Fig. 1D: which cell line is displayed here?

Mut HD1 iPSC is displayed here. We have also revised the figure legend of Fig 1D within Page 1, line 8 to include this information.

(11) Fig. 1E: Karyotype of which cell line is shown?

We have included karyotype of both IsoHD1 and IsoHD2 iPSC in Fig 1E, and also revised the legend within Page 1, line 11, to reflect this change.

(12) Supp. Fig. 1: scale bar information missing.

We thank the reviewer for pointing out this and have revised the legend within Page 1, line 17, to include scale bar information.

(13) Fig. 5: legend for A is missing.

We thank the reviewer for pointing out this and have revised the legend within Page 2, line 91, to include Figure (A) within.

1. Supp. Fig. 3A says 30 days, but only 23 days are shown.

We are sorry for making this inadvertent typo and have since aligned the correct days (31 days) shown within the figure (Supp Fig 3A) and legend (Page 3, line 110, 113), as mentioned in the manuscript.

(15) Supp. Fig. 3C: scale bar length is incorrect.

We did a recheck and are confident that the scale car is of the correct length. The images displaying the respective fluorescent channels are proportionately reduced with respect to the main figure (now Supp. Fig. 3D), and thus are of the same size (200 uM).

(16) Fig. 6: legend for D, E is missing.

We have revised the figure legend within Page 3, line 128, 130 and 131, to address said missing legends.

(17) Stem cells do also express Sox2. how does Sox2 expression lead to the conclusion of an optimal generated organoid?

We thank the reviewer for pointing this out. Sox2 has been defined as a core intrinsic factor for regulating pluripotency (Avilian et al, 2003, Zhang et al, 2014), as well as lineage specifiers to regulate ectodermal differentiation which is crucial in controlling neural initiation and differentiation from iPSC (Zhao et al, 2004, Thomson et al, 2011, Wang et al 2014). Additionally, Sox2 is highly expressed in proliferating neural progenitor cells as documented in previous iterations of cerebral organoids generation protocol (Lancaster et al 2013, Qian at el, 2018). Perhaps “optimally” sounds too forced in this context, as such we have toned down on the phrasing.

(18) HD1 and HD2 react differently (e.g. in IL-1B production), but the text is written often as if both cell lines react in the same way.

We thank the reviewer for pointing this out and have since clarified this within Page 4, line 366-368, of the manuscript.

(19) Precise information on medium missing (e.g. no Pen/Strep?).

We thank the reviewer for pointing out this. Culturing of iPSC colonies was done without the use of Pen/Strep. Additionally, we have elaborated the medium composition for our iMac cultures for clarity within Page 4, line 106, of Materials and Methods as well as the information within Supp. Table 4.

(20) How was ReleSR used exactly?

We have included the usage of ReleSR within Page 2, line 41 of Materials and Methods.

(21) What kind of microscopes/objectives were used for imaging?

We have added the respective microscope details for bright-field, phase-contrast and cytospin related experiments within Page 3, line 73, and Page 14, line 360, of Materials and Methods.

(22) For the dissociation of organoids: what kind of pipit was use and at which temperature were organoids incubated?

We have included the pipette used for organoids dissociation, as well as the incubation temperature for organoids culture within Page 9, line 243, 244 and 245, of Materials and Methods.

(23) How was the RNA-seq analysis done? Which packages? Which versions?

We provide now the information requested in the material and method section.